# DeepSimHO: Stable Pose Estimation for Hand-Object Interaction via Physics Simulation

**Rong Wang**    **Wei Mao**    **Hongdong Li**
The Australian National University
`{rong.wang, wei.mao, hongdong.li}@anu.edu.au`

## Abstract

This paper addresses the task of 3D pose estimation for a hand interacting with an object from a single image observation. When modeling hand-object interaction, previous works mainly exploit proximity cues, while overlooking the *dynamical* nature that the hand must stably grasp the object to counteract gravity and thus preventing the object from slipping or falling. These works fail to leverage dynamical constraints in the estimation and consequently often produce unstable results. Meanwhile, refining unstable configurations with physics-based reasoning remains challenging, both by the complexity of contact dynamics and by the lack of effective and efficient physics inference in the data-driven learning framework. To address both issues, we present *DeepSimHO*: a novel deep-learning pipeline that combines forward physics simulation and backward gradient approximation with a neural network. Specifically, for an initial hand-object pose estimated by a base network, we forward it to a physics simulator to evaluate its stability. However, due to non-smooth contact geometry and penetration, existing differentiable simulators can not provide reliable state gradient. To remedy this, we further introduce a deep network to learn the stability evaluation process from the simulator, while smoothly approximating its gradient and thus enabling effective back-propagation. Extensive experiments show that our method noticeably improves the stability of the estimation and achieves superior efficiency over test-time optimization. The code is available at `https://github.com/rongakowang/DeepSimHO`.

## 1   Introduction

3D hand-object pose estimation from a single image facilities a wide range of applications, including extended reality (XR) [2], human-robot interaction [32], and animation [17]. To this end, it has drawn increasing attention in recent years [34, 52, 56, 20, 19, 5]. A key challenge in this task is to ensure the estimated pose conforms to real-world physics, *i.e.* the hand should make contact with the object without penetrating it, and more importantly, the contact can form a stable grasp to counteract gravity. In this work, we aim to estimate hand-object pose that is both accurate and stable.

To model hand-object interaction, related works [21, 5, 57, 18, 20] often impose distance-based attraction and repulsion constraints that encourage contact and penalize penetration. However, this approach only enforces the proximity and do not explicitly reason the dynamical effects of the contact. In consequence, it allows the hand to simply touch the object, which can not form a stable grasp. As illustrated in Figure 1, such contact can cause the object to slip or fall under gravity, therefore is physically unstable. Meanwhile, in other related tasks such as grasping synthesis [7, 50] and dexterous manipulation [43, 12], physics simulators [48, 8, 54] are commonly used to ensure stable grasping. However, as discussed in [50, 55], existing differentiable simulators supporting mesh-to-mesh contact often produce numerically unreliable state gradient due to the discontinuity in contact geometry [54] and penetration [48], therefore making them difficult to integrate into the

37th Conference on Neural Information Processing Systems (NeurIPS 2023).

learning framework. Alternatively, [51] perform a brute-force search over predefined poses, however, the approach is time-consuming and only constrained to a limited configuration space.

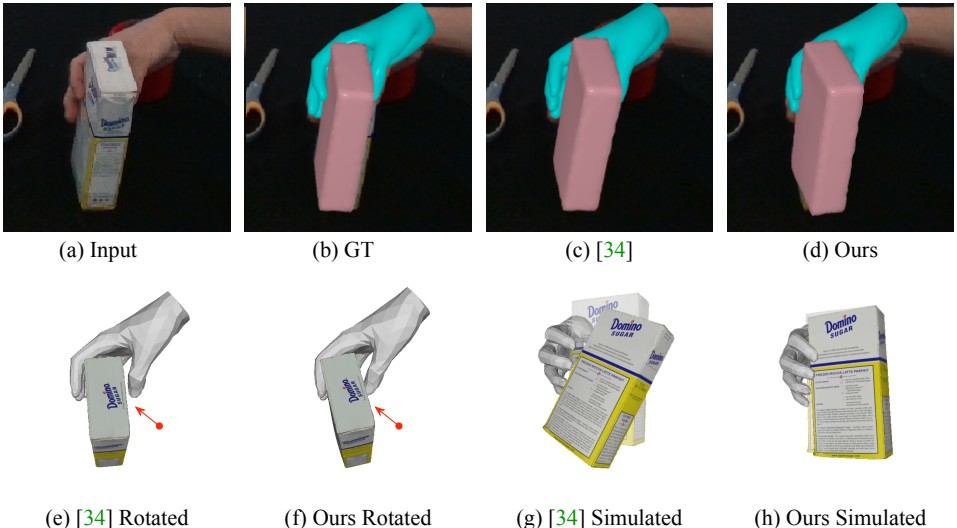

(a) Input    (b) GT    (c) [34]    (d) Ours

(e) [34] Rotated    (f) Ours Rotated    (g) [34] Simulated    (h) Ours Simulated

Figure 1: **Illustration of our method.** Given an input image (a), recent work [34] can estimate accurate hand-object pose that is close to the ground truth (b) in the camera view (c). However, due to occlusion ambiguity and lack of explicit dynamics constraints modeling in hand-objection interaction, they can produce physically unstable results, where the hand only touches the object at one side and thus can not form a stable grasp (e). Such contact can cause the object to fall under gravity (g). In contrast, our result is both accurate (d) and physically stable, as in (f) and (h).

In this paper, we propose an effective and efficient method *DeepSimHO* for estimating accurate and stable hand-object pose, by combining physics simulation in the forward step and gradient approximation with a deep network in the backward step. Specifically, for an initial hand-object pose estimated from a base network, we leverage a physics simulator to evaluate its *stability*, which is measured by the object center displacement due to gravity and hand contact. As shown in Figure 1, optimally stable hand-object pose results in minimal displacement after simulation. However, due to the aforementioned noisy gradient issues, the evaluated stability can not be directly used as a loss to refine the base network via back-propagation. To address this issue, we further introduce a neural network named *DeepSim* to reproduce the stability evaluation and back-propagate through it. On one hand, DeepSim acts as a smooth approximation of the state gradient from the simulator. On the other hand, it *learns from the simulator* to determine the stability of the initial pose, akin to the discriminator in the Generative Adversarial Network (GAN) [16]. However, as there is no adversarial relationship between the base network and DeepSim, the training process is much simpler. In training, we jointly refine the base network with DeepSim in an alternating order similar to [16]. In testing, we can estimate physically more stable poses by a simple forward call of the refined base network, contrasting with [56, 51, 20] that require a computationally expensive test-time optimization.

Our contributions can be summarized as follows. (*i*) We propose a novel pipeline that connects a learning framework with a physics simulator to estimate accurate and stable hand-object pose. (*ii*) We introduce DeepSim, a deep network that learns from simulator for stability evaluation and smooth gradient approximation. Extensive experiments on multiple benchmarks show that our method noticeably improves the stability of the estimation while maintaining comparable accuracy.

## 2 Related Works

**Hand-Object Pose Estimation.** The problem of estimating hand and object poses from monocular images is ill-conditioned due to depth and occlusion ambiguity. To enforce physically plausible results, related works have employed three categories of approaches to model constraints in contact: data-driven [34, 19, 10, 52], distance-based [21, 20, 5, 18] and physics-based refinement [56, 26]. The data-driven approach leverages real [18, 6] or synthetic [34, 21] data to generate natural contact as a result of accurate estimation. Contact correlations can also be implicitly learned from the data using

the attention [19, 49, 52] or graph convolution [10]. However, this approach is sensitive to noise in the annotation, and perfect estimation is difficult to achieve in practice. In contrast, [21, 20, 5] explicitly employ attraction and repulsion constraints using the signed distance field (SDF) to encourage contact and penalize inter-penetration. However, such constraint only enforces the hand and object vertices to be in proximity, regardless of whether the hand can stably grasp the object or not. While additional affinity labels like [3, 27] can mitigate this issue, they require significantly more efforts to annotate.

Recently, physics-based refinement have been widely adopted in many tasks, including 3D shape modeling [36, 37], human motion estimation [13], and particular those for modeling hand-object interaction, such as task-oriented dexterous manipulation [43, 12] and grasp synthesis [7, 50]. In these works, a sequence of control forces is optimized to ensure a stable grasp on the target object, which is typically achieved via deep reinforcement learning. Motivated by their success, several works [51, 56, 26] have applied physics-based refinement for hand-object pose estimation. [51] query in a physics simulator and search for local configurations that are stable in a brute-force fashion. [56] model the contact as a virtual spring-mass system and minimize the elastic energy. [26] compute the acceleration from the initial kinematic estimation over multiple frames, then optimize for the contact force that aligns best with the regressed acceleration. However, all above approaches rely on computationally expensive post-fitting processes and can only handle a limited configuration space, defined by candidate finger parts [51, 26] or anchors [56].

**Physics Simulation.** Physics simulators for modeling contact dynamics between rigid bodies have been widely used in robotics [43, 12, 31, 30]. Recently, differentiable simulators [38, 40, 39] that can compute the state gradient, *i.e.* the derivative of the simulated state with respect to the input state, have drawn increasing attention. However, they suffer intrinsic issues of unstable numerical gradient caused by the non-smoothness in the linear complementary problem (LCP) [55, 50]. In practice, some also have restrictions in applications, such as being limited to primitive body shapes like spheres or cubes [25, 14, 9], which prevent them from being generally applied. Meanwhile, several works [23, 33, 46, 41] have introduced deep neural networks in physics simulations, primarily aimed to reduce the sim-to-real gap [23]. Specifically, [46, 41] adopt a graph convolution network to learn to predict physics states as a replacement or refinement of the simulator, while [23, 33] leverage a hybrid framework that regresses underlying physics quantities for fine-grained control. However, these works require training on a large amount of real-world physics data and thus can not handle noisy configurations, *e.g.* when the hand penetrates the object, since they do not exist in real-world data. In contrast, we introduce the neural network to *learn from the simulator* instead of real-world data, therefore can utilize rich training data obtained during simulation and are capable of analyzing noisy configurations. More importantly, all above works attempt to regress *high-dimension* physics quantities or states at each simulation step, while we only regress a *scalar* stability loss, which effectively reduces the difficulty of regression and enjoys superior generalizability.

## 3 Approach

In this section, we present the proposed pipeline as shown in Figure 2. Given an input image $\mathbf{I} \in \mathbb{R}^{H \times W \times 3}$, we first adopt a base hand-object pose estimation network (base network) $f_b(\cdot)$ to estimate the initial hand mesh $\hat{\mathbf{M}}^h \in \mathbb{R}^{778 \times 3}$ and rigid object pose $\hat{\mathbf{p}}^o = [\hat{\mathbf{R}}^o, \hat{\mathbf{t}}^o] \in SE(3)$ (Section 3.1). Next, we collect the initial configuration $\mathbf{q}_0 = [\hat{\mathbf{M}}^h, \hat{\mathbf{p}}^o]$ and query its stability from a simulator $f_s(\cdot)$ (Section 3.2). To effectively refine the base network, we further introduce the DeepSim network $f_d(\cdot)$ to learn from the simulator and smoothly approximate its gradient (Section 3.3). Finally, we describe the overall training objectives (Section 3.4) to train the proposed model.

### 3.1 Base Hand-Object Pose Estimation Network

**Hand Pose Estimation.** Motivated by previous works [34, 59], we adopt a two-stage approach to estimate the hand mesh $\hat{\mathbf{M}}^h$. In particular, at the first stage, we follow the architecture in [47] to estimate the hand joint locations using heatmaps. Given the input image, we first adopt a ResNet-34 [22] encoder pre-trained on the ImageNet [45] to extract image features. Next, we decode the image feature with deconvolution layers and obtain $K$ hand heatmaps $\hat{\mathbf{H}}^h \in \mathbb{R}^{K \times D \times D \times D}$ after the softmax normalization. Each heatmap $\hat{\mathbf{H}}_k^h \in \mathbb{R}^{D \times D \times D}$ represents the 2.5D location distribution of the $k$-th joint in the discretized pixel and depth spaces with $D$ partitions. The expected value $[u_k, v_k, z_k] \in [0, H] \times [0, W] \times [-R, R]$ of such distribution determines the position of this joint.

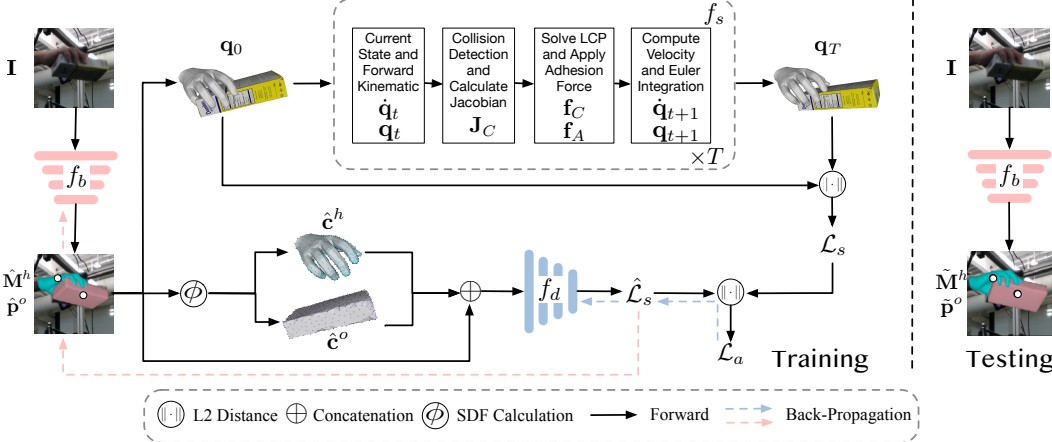

Figure 2: **Overview of our method.** Given an input image $\mathbf{I}$, we aim to enhance the stability of estimated poses from a base network $f_b(\cdot)$ by learning from physics simulation. Specifically, we forward the initial configuration $\mathbf{q}_0$ consisting of the estimated hand mesh $\hat{\mathbf{M}}^h$ and object pose $\hat{\mathbf{p}}^o$ to a physics simulator $f_s(\cdot)$, and evaluate the stability loss as the discrepancy between the simulated configuration $\mathbf{q}_T$. We further use another network named DeepSim $f_d(\cdot)$ to replicate the stability loss value from the same initial configuration, so that the gradient can be effectively back-propagated through it. In testing, only the refined $f_b(\cdot)$ is needed to generate more stable poses $\tilde{\mathbf{M}}^h$ and $\tilde{\mathbf{p}}^o$.

$R > 0$ represents the depth radius relative to the hand root joint and is pre-calculated from the training data following [52]. From the $uvz$ joint positions, we then recover their Euclidean coordinates $\hat{\mathbf{J}}^h \in \mathbb{R}^{K \times 3}$ in the camera frame using the given camera intrinsic $\mathbf{K}$. At the second stage, we employ a pretrained inverse kinematics (IK) network [35] to regress the MANO [44] parameters $\boldsymbol{\theta}$ and $\boldsymbol{\beta}$ from the estimated joint positions. The pose parameter $\boldsymbol{\theta} \in \mathbb{R}^{16 \times 3}$ represents the joint rotations along the hand kinematic tree, including the global rotation. The shape parameter $\boldsymbol{\beta} \in \mathbb{R}^{10}$ represents the hand shape in PCA components. The final hand mesh vertices positions $\hat{\mathbf{M}}^h$ can be differentiably reconstructed from the MANO layer [21].

**Object Pose Estimation.** We follow previous works [52, 34] to assume a known object mesh template $\mathbf{M}^o \in \mathbb{R}^{1000 \times 3}$ in the canonical frame, and predict the rigid object pose $\hat{\mathbf{p}}^o$ that transforms the template into the camera frame. Using the same image feature shared with the hand branch, we adopt a multi-layer perceptron (MLP) to regress the 6D representation [58] of the object rotation $\hat{\mathbf{R}}^o$, and estimate the object center translation $\hat{\mathbf{t}}^o$ relative to the hand root joint in a separate heatmap head. It is worth noting that the simulation-based refinement described in following sections is not limited to the above base network and makes no assumption on its architecture.

## 3.2 Physics Simulation and Stability Loss

**Physics Simulation.** Given the initial configuration $\mathbf{q}_0 = [\hat{\mathbf{M}}^h, \hat{\mathbf{p}}^o]$, we wish to analyze whether it can form a stable grasp or not. To this end, we leverage a physics simulator to evaluate the pose stability considering the dynamical effects of gravity, penetration and contact. Specifically, the physics simulator $f_s(\cdot)$ can be considered as a function that advances the current state, including the configuration $\mathbf{q}_t$ and velocity $\dot{\mathbf{q}}_t$ at time step $t$ as,

$$f_s(\mathbf{q}_t, \dot{\mathbf{q}}_t) = [\mathbf{q}_{t+1}, \dot{\mathbf{q}}_{t+1}], \quad \dot{\mathbf{q}}_0 = \mathbf{0}, \quad t = 0, 1, \cdots T - 1 . \tag{1}$$

The state transition is governed by the Lagrangian dynamics equation [1, 48] with a semi-implicit Euler integration scheme [48], which we formulate for our task as

$$\begin{aligned} \mathbf{M}(\mathbf{q}_t)\dot{\mathbf{q}}_{t+1} &= \mathbf{M}(\mathbf{q}_t)\dot{\mathbf{q}}_t + \mathbf{c}(\mathbf{q}_t, \dot{\mathbf{q}}_t)\Delta t + \mathbf{J}_C(\mathbf{q}_t)^T(\boldsymbol{f}_C(\mathbf{q}_t, \dot{\mathbf{q}}_t) + \boldsymbol{f}_A(\mathbf{q}_t))\Delta t \\ \mathbf{q}_{t+1} &= \mathbf{q}_t + \Delta t \dot{\mathbf{q}}_{t+1} , \end{aligned} \tag{2}$$

where $\mathbf{M}$ represents the hand-object inertia matrix, $\mathbf{c}$ represents the bias forces including the gravitational and Coriolis force, and $\Delta t$ represents the time duration of each step. The hand-object interaction produces the contact force $\boldsymbol{f}_C$ expressed in the contact frame, which consists of the normal force to separate inter-penetrating bodies and the frictional force to resist relative siding. In

addition, human imposes torques on finger joints to counteract gravity when grasping the object. However, as our goal is to refine the stability for the initial hand pose, we require the hand remains static during simulation and thus joint torques are not applicable as it can change the hand pose. To model a similar effect we alternatively introduce an adhesion force $\boldsymbol{f}_A$ in the contact normal direction to maintain attraction to the object. In this way, stable contact can be established without biasing towards deeper penetration as in [56]. We empirically adjust the quantity of $\boldsymbol{f}_A$ to ensure a simple touch is not sufficient to maintain the stability, as shown in Figure 1. Finally, both $\boldsymbol{f}_C$ and $\boldsymbol{f}_A$ are mapped to the generalized coordinate by the contact Jacobian $\mathbf{J}_C(\mathbf{q}_t)$. The overall dynamics equation can be solved numerically via the steps shown in Figure 2. In the following section we describe the inference of stability based on the simulation results.

**Stability Loss.** Motivated by [7], we impose a stability loss as the object center displacement after a period of simulation time $T$ as

$$\mathcal{L}_s = ||\mathbf{t}_T - \mathbf{t}_0||_2, \quad \mathbf{t}_T \in \mathbf{q}_T \quad \mathbf{t}_0 = \hat{\mathbf{t}}^o \in \mathbf{q}_0 . \tag{3}$$

In this formulation, we assume a perfectly stable estimation enables the hand to stably grasp the object, avoiding (*i*) free fall due to being out-of-contact, (*ii*) slipping due to unrealistic interaction, *e.g.* with fingers only touching to the object at one side, and (*iii*) recoil due to penetration resolving by the normal force. All above implausible configurations result in a large displacement and thus are unstable. Our goal is therefore to minimize the stability loss for the initial configuration $\mathbf{q}_0$, which can be achieved by refining the base network via back-propagation.

However, existing differentiable simulators can not provide reliable gradient for computing $\partial\mathcal{L}_s/\partial\mathbf{q}_0$, making the back-propagation intractable through the simulator. First, the contact normal of object meshes are discontinuous at corners of triangle faces, thus the analytical gradient of the normal force can be undefined at these singular points [50]. Second, simulators like [48] often apply a large normal force to quickly resolve the penetration, which results in a large gradient for small pose perturbation in numerical gradient methods. Consequently, direct gradient calculation can be noisy and does not benefit the refinement, as will be shown in Figure 4. To address this issue, we propose the *DeepSim* network which acts as a smooth approximation of the gradient as described in the following section.

## 3.3 DeepSim Network

To smoothly approximate the stability evaluation process, we introduce a differentiable function that learns to produce the same stability loss as the simulator will given any initial configuration. To this end, we parameterize such function with a MLP named *DeepSim*. Specifically, since the simulator solves for the contact force based on the contact points, we first compute two feature vectors $\hat{\mathbf{c}}^h \in \mathbb{R}^{778}$ and $\hat{\mathbf{c}}^o \in \mathbb{R}^{1000}$ from the initial configuration, which represent the signed distances of hand vertices to the object mesh and object vertices to the hand mesh respectively. The two vectors jointly provide rich local contact information to facilitate the stability regression. We then concatenate the two vectors with the flattened initial configuration as the input for the DeepSim $f_d(\cdot)$ and regress the replicated stability loss $\hat{\mathcal{L}}_s$ as

$$\hat{\mathcal{L}}_s = f_d(\hat{\mathbf{M}}^h \oplus (\hat{\mathbf{R}}^o\mathbf{M}^o + \hat{\mathbf{t}}^o) \oplus \hat{\mathbf{c}}^h \oplus \hat{\mathbf{c}}^o) , \tag{4}$$

where $\oplus$ represents concatenation. The objective of the DeepSim network is therefore to minimize the approximation loss as

$$\mathcal{L}_a = ||\hat{\mathcal{L}}_s - \mathcal{L}_s||_2 . \tag{5}$$

In contrast to [46, 41] that regress all components in the simulated state at each step, we directly predict the final stability loss as a scalar, which is empirically an easier objective and better generalizes to noisy unstable configurations, as will be compared in Section 4.5.

**Joint Training and Refinement.** Similar to the GAN [16], we jointly train the base network and DeepSim in an alternating order for an effective refinement. When updating the DeepSim, we use the approximation loss as in Eq.(5) while fixing the base network weights. As the training progresses, the base network produces various configurations with corresponding stability losses automatically labelled by the simulator, which serve as rich training pairs for the DeepSim. In addition, to avoid the DeepSim overfitting to the base network outputs, we perform random perturbations in the estimated hand and object poses as data augmentation. We further use ground truth poses to train the DeepSim in order to include more data other than the base network output. Meanwhile, when updating the base

network, DeepSim facilities to back-propagate the smoothed gradient with respect to the replicated stability loss, with its own weights fixed. Note that we freeze weights in the shared image encoder when refining with the stability loss to avoid corruption of image features. Finally, to avoid being misled by an incorrect regression in the DeepSim, we mask out the replicated stability loss on training samples where $\hat{\mathcal{L}}_s$ and $\mathcal{L}_s$ do not fall in the same range. More details are discussed in the supplementary materials.

## 3.4 Training Objectives

To ensure the estimated poses are both stable and accurate, we impose a multi-task training objective for the base network, which contains two components: the replicated stability loss $\hat{\mathcal{L}}_s$ defined in Eq.(4) and accuracy losses as described in the following. Specifically, we first adopt an $L_2$ loss to supervise the hand joint and mesh estimation as

$$\mathcal{L}_h = ||\hat{\mathbf{J}}^h - \mathbf{J}^h|| + ||\hat{\mathbf{M}}^h - \mathbf{M}^h|| , \tag{6}$$

where $\mathbf{J}^h$ and $\mathbf{M}^h$ represent ground truth hand joints and mesh vertices positions respectively. Next, we impose an object loss to supervise the object pose estimation as

$$\mathcal{L}_{o_1} = ||(\hat{\mathbf{R}}^o \boldsymbol{c}^o + \hat{\mathbf{t}}^o) - (\mathbf{R}^o \boldsymbol{c}^o + \mathbf{t}^o)|| , \tag{7}$$

where $\boldsymbol{c}^o$ represents the 8 tightest object corners positions in the canonical frame, and $[\mathbf{R}^o, \mathbf{t}^o]$ represents the ground truth object pose. To account for the object rotation symmetry, we follow [19] to impose a symmetry-aware object loss as

$$\mathcal{L}_{o_2} = \min_{\mathbf{R}^o \in \mathcal{S}} ||(\hat{\mathbf{R}}^o \boldsymbol{c}^o + \hat{\mathbf{t}}^o) - (\mathbf{R}^o \boldsymbol{c}^o + \mathbf{t}^o)|| , \tag{8}$$

where $\mathcal{S}$ represents the set of all equivalent rotation matrices defined by the objects' symmetry axes [4]. Furthermore, we borrow from [34] to impose an ordinal loss $\mathcal{L}_d$ that supervises for the coarse hand-object relative position. The overall loss is a weighted sum of all individual loss functions as

$$\mathcal{L} = \lambda_h \mathcal{L}_h + \lambda_{o_1} \mathcal{L}_{o_1} + \lambda_{o_2} \mathcal{L}_{o_2} + \lambda_d \mathcal{L}_d + \lambda_s \hat{\mathcal{L}}_s , \tag{9}$$

where $\lambda_h, \lambda_{o_1}, \lambda_{o_2}, \lambda_d, \lambda_s$ are hyper-parameters.

# 4 Experiments

## 4.1 Datasets

We evaluate our method and state-of-the-art methods on two datasets: DexYCB [6] and HO3D [18].

**DexYCB.** The DexYCB dataset [6] consists of 582K frames involving 20 different YCB objects [4]. We use the official "S0" train-test split for the training and evaluation. Following [52], we evaluate on right hand poses and filter out samples in which the hand or object is not within the field of view of the camera. To ensure consistent comparison in physics metrics, we remove test samples where the hand does not interact with the object and only select those that remain stable after simulation (see the GT results in Table 1), resulting in a total of 6348 samples. For training, we do not perform this selection of stability in order to include more data. However, we mask out the replicated stability loss on unstable training samples, *e.g.* no hand-object interaction, to avoid misleading supervision.

**HO3D.** The HO3D dataset [18] consists of 66K frames featuring 10 different objects. We select the "v2" version that is mostly evaluated by previous works [56, 20, 52, 34]. Since its ground truth hand poses for the test set are not released, we follow [56] to evaluate on a subset named "v2$^-$", whose physics plausibility is manually verified by [56]. For training data, we use the official HO3D v2 training split and follow the same practice as the DexYCB dataset to perform sample selection and loss masking. The total HO3Dv2$^-$ test set consists of 6076 samples.

## 4.2 Metrics

**Hand & Object Metrics.** We evaluate the accuracy of pose estimation by comparing the hand and object results with the ground truth labels. For the hand metric, we calculate the mean joint error

(MJE) [60], which measures the Euclidean distance between the predicted and ground truth hand joint positions after root joint alignment. For the object metric,we compute the mean corner error (MCE) on the HO3D test set following [34], which calculates the average distance between the predicted and ground truth positions of the 8 tightest object corners. Since on the DexYCB dataset, many objects are rotationally symmetric, we thus borrow from [19] to compute the symmetry-aware mean corner error (SMCE) as defined in Eq.(8), for a fair comparison with [34].

Table 1: **Quantitative comparison on the DexYCB test set.** Best results are highlighted in **bold** and and inapplicable results are marked with "-". Our method achieves superior physics plausibility and stability with comparable accuracy compared to state-of-the-art methods [20, 52, 34].

| | Hand | Object | Physics | | | |
|---|---|---|---|---|---|---|
| Methods | MJE $(cm)\downarrow$ | SMCE $(cm)\downarrow$ | CP $(\%)\uparrow$ | PD $(cm)\downarrow$ | SD $(cm)\downarrow$ | SR $(\%)\uparrow$ |
| GT | - | - | 100 | 0.91 | 0.64 | 100 |
| Hasson *et al.* [20] | 1.25 | - | 84.35 | 1.80 | 4.83 | 16.80 |
| ArtiBoost [34] | **1.07** | **1.60** | 94.23 | 1.50 | 2.78 | 30.07 |
| Wang *et al.* [52] | 1.15 | - | 89.16 | 1.57 | 3.53 | 19.54 |
| Ours | 1.12 | 1.73 | **95.90** | **1.48** | **2.42** | **32.89** |

Table 2: **Quantitative comparison on the HO3Dv2$^-$ test set.** Best results are highlighted in **bold** and and inapplicable results are marked with "-". Our method consistently outperforms baseline methods [20, 52, 34, 56] in all physics metrics while maintaining comparable accuracy.

| | Hand | Object | Physics | | | |
|---|---|---|---|---|---|---|
| Methods | MJE $(cm)\downarrow$ | MCE $(cm)\downarrow$ | CP $(\%)\uparrow$ | PD $(cm)\downarrow$ | SD $(cm)\downarrow$ | SR $(\%)\uparrow$ |
| Hasson *et al.* [20] | - | 5.35 | 78.52 | 2.02 | 6.40 | 10.37 |
| Yang *et al.* [56] | - | 5.74 | 96.47 | 1.65 | 3.16 | 17.00 |
| ArtiBoost [34] | - | 4.86 | 94.47 | 1.27 | 2.83 | 18.02 |
| Wang *et al.* [52] | - | **4.79** | 93.07 | 1.88 | 3.47 | 11.42 |
| Ours | - | 5.28 | **96.64** | **1.17** | **2.42** | **19.24** |

**Physics Metrics.** We evaluate the physical plausibility of the estimation following the criterions in [7, 56, 20]. Specifically, we evaluate the simulation displacement (SD) [56] to compute the average object center displacement after 200ms simulation time. We then evaluate the success rate (SR) [7] to classify the prediction whose SD is below a threshold of 1cm as success. These two metrics are used to evaluate the pose stability. In addition, we evaluate the contact percentage (CP) [28] as the ratio of predictions with hand-object contact and the penetration depth (PD) [3] as the maximum penetration distance of contacting hand-object predictions. It is worth noting that a stable estimation, *i.e.* with a lower SD, generally indicates a higher CP and a lower PD, but the converse is not necessarily true.

### 4.3 Implementation Details

We implement the model in PyTorch [42] and train it using the Adam [29] optimizer on a single NVIDIA RTX 3090 GPU. We adopt the base network pretrained in [34], and independently train the DeepSim for 40 epochs as a warm-up. We then jointly train both networks enabling the stability loss for 25 epochs. We set the learning rate to $5 \times 10^{-5}$. When training on both datasets, we follow [34, 52] to crop input images into $224 \times 224$ pixels using the provided hand-object bounding boxes. In addition, we follow [34] and perform data augmentation with random translation and rescaling by a factor of 0.1. However, we exclude rotation augmentation as it can affect the ground truth stability. Finally, we set $\lambda_h = 0.5, \lambda_d = 0.1, \lambda_s = 0.1$, and follow [34] to set $\lambda_{o_1} = 0, \lambda_{o_2} = 0.2$ on the DexYCB dataset and $\lambda_{o_1} = 0.2, \lambda_{o_2} = 0$ on the HO3D dataset for a fair comparison.

For physics simulation, we use the MuJoCo [48] simulator. We initialize the hand and object inertia matrix $\mathbf{M}$ using the Composite Rigid Body algorithm [11]. We further convexly decompose hand meshes by the MANO blending weights [44] and object meshes by the CoACD [53] algorithm to facilitate continuous collision detection [15] in MuJoCo. We set the gravity acceleration as 9.8 m/s$^2$ in the $y$ direction of the camera frame. For the adhesion force, we empirically set the gain as 100 and the maximum control range as 10, so that a simple touch is not sufficient to stably grasp the object. Finally, we set the simulation step to be $T = 100$ and time duration in each step as $\Delta t = 0.02$. More implementation and training details can be found in the supplementary materials.

## 4.4 Results

**Quantitative results.** We compare our method with three categories of state-of-the-art methods: data-driven [34, 52], distance-based [20] and physics-based [56] approaches. In Table 1, we report the comparison on the DexYCB test set and re-evaluate all baseline methods using the officially provided weights [34, 52] or optimization scripts [20, 56]. To avoid unfair comparison, we exclude the SMCE result for [52, 20] as they do not consider object symmetry in implementation. From the table, we observe that our method achieves superior performance on all physics metrics, while maintaining comparable accuracy using significantly less data than [34, 52] when trained with the stability loss. Our method also noticeably improves the stability compared to distance-based approach [20].

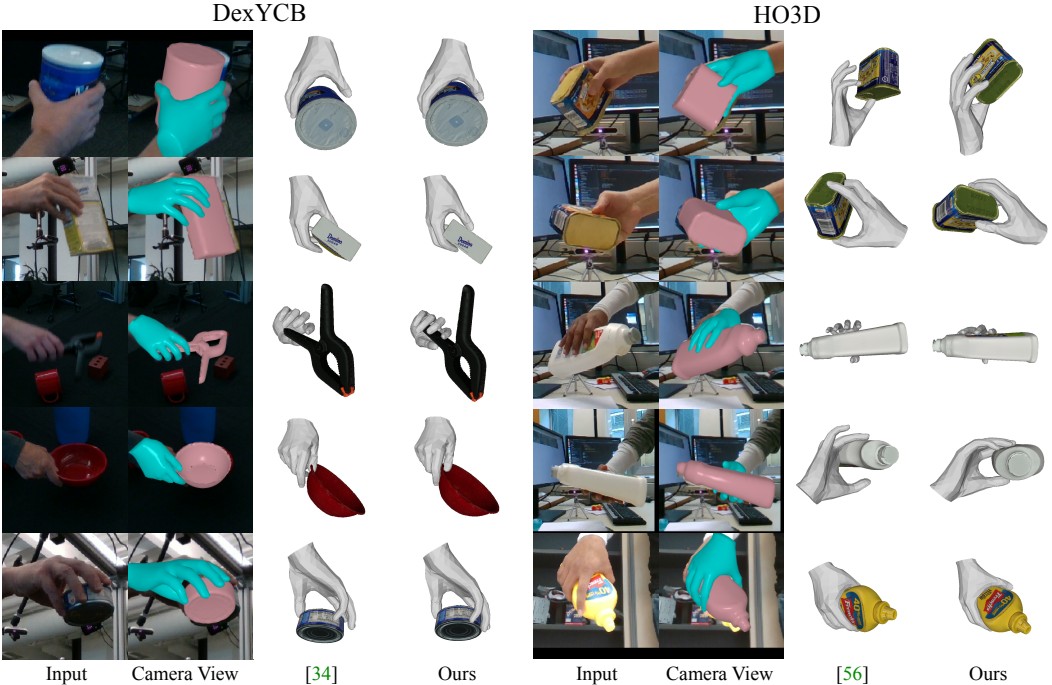

Figure 3: **Qualitative results on the HO3D and DexYCB test sets.** State-of-the-art methods [34, 56] produce results that are either unstable, or stable but do not align with the input image, *e.g.* last row in the HO3D. In contrast, our method produces poses that are both stable and comparably accurate.

In Table 2, we compare all methods on the HO3Dv2⁻ test set. Since the selected split is not suitable for submitting to the official HO3D online evaluation server, we follow the same evaluation protocol as [56] to not evaluate for the hand metric. In addition, although [56] also report the SD in their paper, they use a different simulator [8] and the code for this part is not publicly available. Hence we re-evaluate the same metric using our physics modeling for a fair comparison. Consistently, we observe our method noticeably improves physics plausibility and stability with comparable accuracy. Finally, for the testing, [56] and [20] requires 12 and 45 seconds to optimize for one input image on a single NVIDIA RTX 3090 GPU, in comparison, our method only requires 0.02 seconds on the same device for a forward call of the refined base network, which is significantly more efficient.

**Qualitative results.** We present qualitative results on the DexYCB and HO3D test sets in Figure 3. We render the refined hand and object meshes in the camera view, illustrating that the estimated poses align well with the input image. More importantly, we compare in rotated views with [34, 56] to show that while [34] produce visually close results, the established contact is actually physically unstable, with some fingers floating in the air or penetrating the object. In addition, [56] do not consider pose accuracy when optimize for physics plausibility, therefore can generate results that are stable but do not align with the input image. In comparison, our results are both accurate and physically stable.

## 4.5 Ablation Study

**Effects of the DeepSim Network.** To verify the effects of the DeepSim for back-propagation, we compare with two other approaches for calculating the state gradient: (*i*) numerical gradient using

the finite difference in the official MuJoCo implementation, and (*ii*) analytical gradient through the differentiable simulator NimblePhysics [54]. We use a similar physics model in the NimblePhysics based on its available functions (included in the supplementary materials). To avoid training failure, we include a gradient clip of 1.0 for all methods. In Figure 4 (b), we observe (*i*) produces incorrectly large gradient due to the sudden velocity change during penetration resolving. For (*ii*), we observe it

Table 3: **Comparison on design choices of DeepSim and stability loss.** Best performing method (MLP+S) directly regresses a final scalar stability loss, which tackles a easier task and therefore outperforms variants that regress more state components (T/RT) or more simulation steps (LSTM).

| Methods | DeepSim
AE ($mm$)↓ | Physics
CP (%)↑ | PD ($cm$)↓ | SD ($cm$)↓ | SR (%)↑ |
|---|---|---|---|---|---|
| w/o DeepSim | - | 94.19 | 1.51 | 2.77 | 28.55 |
| MLP + T | 13.80 | 94.16 | 1.53 | 2.76 | 28.51 |
| MLP + RT | 17.22 | 94.57 | 1.52 | 2.81 | 26.60 |
| LSTM + T | 10.79 | 94.73 | 1.50 | 2.60 | 30.66 |
| MLP + S | **2.70** | **95.90** | **1.48** | **2.42** | **32.89** |

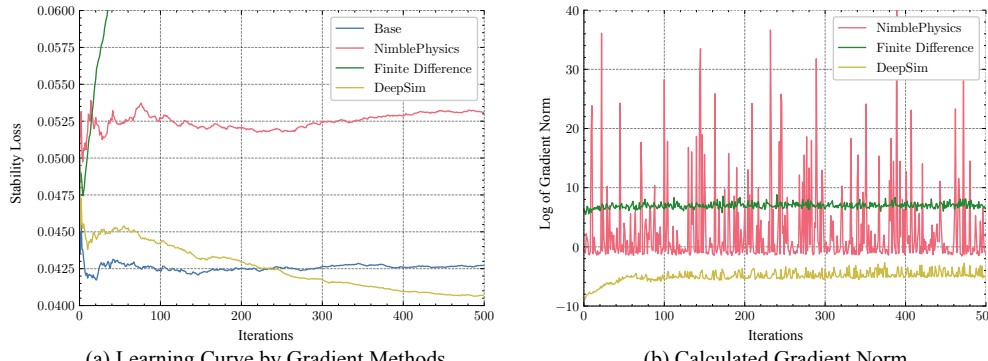

(a) Learning Curve by Gradient Methods      (b) Calculated Gradient Norm

Figure 4: **Effects of the DeepSim for back-propagation.** Figure (a) shows the trend of the stability loss (evaluated from the MuJoCo simulator) as the training progresses. Figure (b) shows the $\log$ of the raw gradient norm at each iteration ($||\partial \mathcal{L}_s / \partial \mathbf{q_0}||$ for analytical gradient (NimblePhysics) and numerical gradient (Finite Difference), and $||\partial \hat{\mathcal{L}}_s / \partial \mathbf{q_0}||$ for the DeepSim). Gradients directly obtained from the simulators are not suitable for training. In contrast, DeepSim provides numerically stable gradient and benefits the refinement over Base method trained with accuracy losses only. generates numerically unstable gradient, and even causes numerical overflow at singular points due to the discontinuity in contact geometry. The noisy gradient prevents the base network from being effectively refined as shown in Figure 4 (a). In contrast, the gradient through DeepSim is numerically stable and can better benefit the refinement over training with accuracy losses only.

**Design Choices.** In Table 3, we compare different variants of design in DeepSim and stability loss. We observe that regressing more state components, *e.g.* the 3 DoF. object center (MLP + T) or 6 DoF. object pose (MLP + RT) in the final simulation step, can be more challenging compared to regressing a scalar stability loss (MLP + S). As a result the DeepSim performs worse with a higher approximation error (AE), which measures the discrepancy between $\hat{\mathcal{L}}_s$ and $\mathcal{L}_s$. Underfitting in DeepSim can cause severe misleading, thus can not help to improve the stability compared to the baseline method that only trains with accuracy losses (w/o DeepSim). In addition, we replace the MLP with a long short-term memory network (LSTM) [24] (implementation details in the supplementary materials), which sequentially estimates residual center displacements in each step (LSTM + T), so that the overall stability loss is the norm of the total displacement. However, we observe that dividing the total displacement into multiple partitions can cause the DeepSim to overfit by predicting only small residuals, therefore does not outperform against directly regressing for the final simulation step.

## 5 Discussion

**Limitations & Societal Impacts.** Although DeepSim addresses the issues of simulator gradient in the back-propagation, our physics modeling and simulation in the forward step are still dependent on the simulator implementation, which can be imperfect when compared to real-world physics. In addition, we made several restrained assumptions about the physics properties and contact dynamics,

for instance we assumed rigid objects with known gravity direction, as well as simplified modeling of joint torques via the adhesion force. As a result, the potential sim-to-real gap may lead to unsafety when interacting with objects that have complex and unknown physics properties.

**Future Works.** The proposed pipeline can be benefited with improved designs of physics metrics, simulator and the DeepSim network. Future works are thus encouraged to explore on (*i*) stability losses with a more comprehensive consideration of the object state, *e.g.* rotation, velocity, (*ii*) more realistic simulation, especially for static hand, (*iii*) variant DeepSim networks that explicitly utilize physics priors for the regression, and (*iv*) extension to video inputs with known object initial states.

**Conclusion.** In this paper, we propose a novel pipeline *DeepSimHO* for estimating accurate and stable hand-object pose. We explicitly model the dynamical nature in hand-object interaction via forward physics simulation and stability evaluation. In the backward step, we introduce DeepSim, a neural network that learns the stability evaluation from the simulation and performs smooth gradient approximation to facilitate refinement of the base network. Thanks to them, we our method effectively and efficiently produces physically stable hand-object pose that conforms better to reality.

## Acknowledgement

The research is funded in part by an ARC Discovery Grant (grant ID: DP220100800) to HL.

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
