# OpenReview forum: "DeepSimHO: Stable Pose Estimation for Hand-Object Interaction via Physics Simulation"
_NeurIPS.cc/2023/Conference — NeurIPS 2023 poster_

### Official Review · Reviewer_fSiU · 2023-07-03

**Soundness:** 3 good
**Presentation:** 3 good
**Contribution:** 2 fair
**Rating:** 5
**Confidence:** 4

**Summary:**

The paper presents a stable hand pose estimation method leveraging a neural network for stability estimation trained from simulation to improve the physical stability of estimated hand poses. The main idea is training a neural network via simulated results. The network can further provide smooth gradients to refine estimated hand poses. Experiments on two datasets demonstrate the effectiveness of the method and the ability of DeepSim network to provide better gradients where the overall learning framework can benefit from.

**Strengths:**

- The idea of training a neural network as a simulator that can both provide accurate simulation results regrading stability and provide smooth gradients friendly for network training is sound. Gradients provided by the network are also demonstrated to be of higher quality compared to analytical gradients or those from finite differences.
- Experiments are valid and reasonable that are able to demonstrate the superiority of the proposed method. Implementation details are also provided.

**Weaknesses:**

- The idea of designing neural networks as a differentiable simulator is not a new thing [1,2]. The network designed in the paper does not has explicit physics priors and only approximates the stability value, which make its generalization ability towards unseen and out-of-distribution data ambiguous.
- Using networks to learn the stability prediction process is not fully explored. For instance, could the current strategy generalize well towards out-of-distribution test data? Is it possible to improve the generalization ability and train a very powerful stability prediction network by creating and leveraging a large scale synthetic dataset via simulator? Is there any opportunity to fuse physical priors into the design of DeepSim for generalization enhancement?



[1] Mezghanni, M., Bodrito, T., Boulkenafed, M., & Ovsjanikov, M. (2022). Physical simulation layer for accurate 3d modeling. In *Proceedings of the IEEE/CVF Conference on Computer Vision and Pattern Recognition* (pp. 13514-13523).

[2] Mezghanni, M., Boulkenafed, M., Lieutier, A., & Ovsjanikov, M. (2021). Physically-aware generative network for 3d shape modeling. In *Proceedings of the IEEE/CVF Conference on Computer Vision and Pattern Recognition* (pp. 9330-9341).

**Questions:**

- Is there any in-depth analysis of the DeepSim network (please see weakness for details)?

- Some works in robotics propose to relax and improve contact models so that smooth gradients can be provided for optimization [1,2]. It is hard to compare with them directly since they are not open-sourced. However, is it possible to conduct toy analysis to compare the effectiveness of soften contact models and the neural simulator proposed in the paper?



[1] Pang, T., & Tedrake, R. (2021, May). A convex quasistatic time-stepping scheme for rigid multibody systems with contact and friction. In *2021 IEEE International Conference on Robotics and Automation (ICRA)* (pp. 6614-6620). IEEE.

[2] Jain, S., & Liu, C. K. (2011, December). Controlling physics-based characters using soft contacts. In *Proceedings of the 2011 SIGGRAPH Asia Conference* (pp. 1-10).

**Limitations:**

Limitations are stated in Section 5.

---

> ### Author Rebuttal · Authors · 2023-08-07
>
> Dear Reviewer fSiU:
>
> We thank you for providing valuable feedback and acknowledging the strength of our work. We hope the below responses can address your concerns.
>
> ### 1. Difference to Previous Physics-based Works.
> We thank the reviewer for providing other related works. However, the suggested works [B4, B5], similar to other works in grasp synthesis [6, 46], all address the task of *synthesizing* physically valid results, which are not directly comparable to our method **as we address a more challenging task of jointly estimating *stable* and *accurate* hand-object poses *conditioned on a monocular image observation***. In addition, **we tackle the problem of effective joint training of the DeepSim and base network to avoid overfitting and enforce the generalizability, which is not addressed by previous works.** For completeness, we will include [B4, B5] in the discussion of related works in the revised paper.
>
> We also refer the reviewer to the general responses #2. for more discussion and insights about our method compared previous works.
>
> ### 2. Generalizability of the DeepSim Network
> The trained DeepSim network can generalize to unseen test data thanks to our training strategy using large-scale perturbed hand-object data. Specifically, We mentioned in L189 of the main paper that to avoid overfitting, we include randomly perturbed initial hand and object poses before forwarding to the simulator when training the DeepSim network (note the augmentation is for training the DeepSim only). **This is essentially the same as generating large-scale synthetic pose-stability data pairs as the training progresses**. In addition, ablation study in Table 3 of the main paper further quantitatively proves the generalizability of the DeepSim network. In Table 3, we observe that the Approximation Error (AE, measuring the distance between the predicted stability and ground truth stability evaluated by the simulator on unseen test data), is significantly smaller than the stability threshold used in the physics metric, **indicating that the DeepSim is sufficiently accurate to distinguish if the estimated hand-object pose is stable or not during testing**, which is the key to the success of our method.
>
> ### 3. Integrating Physics Priors to the DeepSim Network
> In addition to initial hand and object poses, we included two feature vectors $\hat{\mathbf{c}}^h$ and $\hat{\mathbf{c}}^o$ as the contact priors for the input of the DeepSim network to improve its effectiveness, as mentioned in L174 of the main paper. Since the stability of the estimated hand-object pose depends primarily on the contact force, which is further dependent on the contact configuration, *i.e.* contact points, penetration volume etc., **these two vectors embed rich physics priors and help to improve the precision of stability prediction**.
>
> There are various ways to fuse additional physics priors to better faciliate the DeepSim network. First, the DeepSim network can condition on other physics properties, *e.g.* mass, gravity etc. for more accurate regression. In addition, since collision detection and penetration resolving often cause numerically unstable gradient, the DeepSim network can instead aproximate these subroutines by regressing components of contact forces *e.g.* contact positions, penetration direction and volume etc., leaving other rountines like object acceleration and velocity calculation handled by the simulator. We believe these practices are suitable for future research works.
>
> ### 4. Analysis on Soften Contact Model
> We thank the reviewer for providing additional works on the improved soften contact model. However, since we need to compute the overall gradient $\frac{\partial \mathcal{L}_s}{\partial \mathbf{q}_0}$ in order to refine the base network in an end-to-end fashion, **having only improved soften contact model is not sufficient to guarantee the robustness of the overall gradient**. First, as the base network can potentially generate initial poses where the hand penetrates the object, the penetration must be first resolved by the simulator by applying a large normal force to separate the hand and object mesh. We observe numerically unstable gradient often arises at this stage due to the sudden velocity change in penetration resolving. Furthermore, existing methods that successfully leverage differentiable physics simulators mostly consider primitive shapes only, where the collision detection procedure is straightforward. However, in our task the contacting object meshes often have complex and discontinuous mesh geometry, causing the state gradient associated to the collision detection to be numerically unstable as well. Therefore, we believe adopting only a soften contact model can not fully address the gradient issue and emphasize the necessity of the proposed DeepSim network.
>
> ### Bibliography
> [B4] Mezghanni, M., Bodrito, T., Boulkenafed, M., & Ovsjanikov, M. (2022). Physical simulation layer for accurate 3d modeling. In Proceedings of the IEEE/CVF Conference on Computer Vision and Pattern Recognition (pp. 13514-13523).
>
> [B5] Mezghanni, M., Boulkenafed, M., Lieutier, A., & Ovsjanikov, M. (2021). Physically-aware generative network for 3d shape modeling. In Proceedings of the IEEE/CVF Conference on Computer Vision and Pattern Recognition (pp. 9330-9341).

---

> > ### Comment · Reviewer_fSiU · 2023-08-19
> > **Thanks for the rebuttal**
> >
> > Thanks for your clarification. After reading the rebuttal and other reviewers' comments, I think the paper is of some value to be published somewhere. But I still have some concerns about its significance and the potential impact. In short, the paper does not conduct a deep and insightful discussion on an interesting and valuable problem.
> >
> > **The DeepSim network:** The network and the representations used are not designed under careful thinkings or after rigorious explorations. There are many works on learning simulation that are carefully calibrated to inject priors to the network architecure like [PINNs,NCLaw]. In this work, the authors are expected to conduct thorough discussions w.r.t. how to design a network to predict the physical stability of a grasping pose. It involves the netowrk architecture, input and output, what to predict, and so on. Presenting the designing process to readers either in the ablation study or in the supp would make the work more inspring to others. The current content in the paper cannot fully convince readers that the MLP structure is the most suitable one for the stability regression. Seemingly it is not designed under careful considerations. Besides, how to represent the grasping is expected to be discussed in depth. The current approach simply leveraging signed distances from hand to object and from object to hand. An intuitive illustration through figures to demonstrate it's effectiveness in representing the grasping (sometimes are incorrect with penetrations) would be helpful to let others get its insights.
> >
> > **The simulator:** Is the current simulator enough to calculate provide correct simulation? Mujoco was published in 2012. After that, simulators for graphics or robotics have undergone a fast development. A comparison is presented in [Dojo]. Besides, RK4 is used as the integrator in Mojuco. However, it seems that explicit Euler is used here (Eq. 2)? Have you tried other simulators? A discussion w.r.t. which offline simulator should be leveraged to provide GT stability scores is expected to be covered in the paper as well.
> >
> >
> >
> > [PINNs] Raissi, M., Perdikaris, P., & Karniadakis, G. E. (2019). Physics-informed neural networks: A deep learning framework for solving forward and inverse problems involving nonlinear partial differential equations. *Journal of Computational physics*, *378*, 686-707.
> >
> > [NCLaw] Ma, P., Chen, P. Y., Deng, B., Tenenbaum, J. B., Du, T., Gan, C., & Matusik, W. (2023). Learning Neural Constitutive Laws From Motion Observations for Generalizable PDE Dynamics. *arXiv preprint arXiv:2304.14369*.
> >
> > [Dojo] Howell, T. A., Le Cleac’h, S., Kolter, J. Z., Schwager, M., & Manchester, Z. (2022). Dojo: A differentiable simulator for robotics. *arXiv preprint arXiv:2203.00806*, *9*.

---

> > > ### Author Response · Authors · 2023-08-19
> > >
> > > Dear Reviewer fSiU:
> > >
> > > We thank you for acknowledging the value of our paper. We hope the below responses can address your further concerns.
> > >
> > > ### 1. The DeepSim network
> > >
> > > We acknowledge that the design of the DeepSim network is important, however, we wish to highlight that in this paper, **our main focus is about effective and efficient learning from physics simulation to improve the stability of hand-object pose estimation, instead of proposing novel network designs**. Moreover, since the DeepSim network tackles a *simplified regression task* instead of directly approximating the entire simulation process, **we observe that the proposed designs are sufficiently accurate to achieve the goal, as justified in the Table 3 of the main paper**. To further prove the effectiveness, we have compared with several variants on the design choices in the ablation study, including architectures (MLP/LSTM) and predictions (S/T/RT), we will revise and provide more discussions on this in the final version. Finally, we acknowledge that our current designs can be potentially refined in future works to further improve the overall performance, however, exhausting all design variants is impractical and besides the main point of the paper.
> > >
> > >
> > > As we mentioned in the rebuttal, motivated by the simulation process, we include the signed distance vectors to reflect the initial contact configuration and better facilitate the regression. We thank you for your suggestion and will add additional figure illustration on this to clarify the insight in the final version.
> > >
> > >
> > > ### 2. The Simulator
> > > While MuJoCo was first published in 2012, its codebase [B6] is constantly maintained even nowadays to provide improved simulation precision. In addition, it is commonly applied in related works, *e.g.* [B7], considering its robustness and efficiency in collision detection and contact-related simulation. We therefore follow previous works to apply the MuJoCo simulator and empirically observe that the simulation reasonably aligns with real world physics.
> > >
> > > We mentioned in L88 of the main paper that while many other differentiable simulators like [25,12,8] exist, including the [Dojo], they are still on the development and currently only support for gradient calculation of *primitive collision shapes*, which are not applicable to our task since we tackle objects with complex *mesh collision shapes*, where the graident with respect to contact geometry is often problematic.
> > >
> > > The Eq. (2) uses a commonly adopted *implicit* Euler integration scheme, which is the default setting in MuJoCo. Please note that MuJoCo provides different options of solvers, as mentioned in [44] and implemented in [B6].
> > >
> > > We have also tested the NimblePhysics [50] simulator, which is a recent feature-complete differentiable simulator that supports mesh contact shapes. The ablation study shows that the DeepSim network produces gradient of higher quality and better facilitates the back-propagation.
> > >
> > > All GT results are obtained using the same setting mentioned in the implementation details with MuJoCo, we will clarify this in the final version.
> > >
> > > ### Bibliography
> > >
> > > [B6] https://github.com/deepmind/mujoco
> > >
> > > [B7] Dasari, Sudeep, Abhinav Gupta, and Vikash Kumar. "Learning dexterous manipulation from exemplar object trajectories and pre-grasps." 2023 IEEE International Conference on Robotics and Automation (ICRA). IEEE, 2023.

---

> > > > ### Comment · Reviewer_fSiU · 2023-08-19
> > > >
> > > > Thanks for your clarification.
> > > >
> > > > - I do not think changing the backbone (LSTM v.s. MLP) is an exploration towards the network architecture.
> > > > - Regressing the stability score limits the network for replacing simulation to simple grasping scenarios. I do not think it has good scalability and is what we should take if we want to handle more complex and diverse interactions. Therefore, it makes the NN an engineering adhoc design, lacking high impacts or potential future values.
> > > > - Difftaichi is not a simulator but provides examples on how to use the differentiable Taichi language to write simulators. Then, what it can support depends on how you implement your simulator.
> > > > - Besides, when saying the offline simulator, I talk about simulator but not differentiable simulator. The differential ability needs not treated as an important thing when selecting the offline simulator since you rely on it to provide GT stability values but not its gradients. Simulators for robotics commonly only support contacts with primitive shapes. But many simulation algorithms developed in graphics for rigid bodies such as Affine Body Dynamics, Rigid-IPC can do a lot of things, like supporting general mesh geometry, intersection free trajectories. It seems that they can give more accurate simulation. Thus offer accurate stability values. Mujoco requires small timestep due to its integration scheme. That might limit the work to simple grasping other than more complex interactions.
> > > > - I do not have very deep background in simulation. But Equation 2 seems not an implicit integration. It is semi-implicit since $\dot{q}_{t+1}$ only depends on values from the previous timestep. And the integration does not need solving optimization problems. (LCP solves the optimization for calculating contact related values. Here I talk about the time integration.)

---

> > > > > ### Author Response · Authors · 2023-08-20
> > > > >
> > > > > Dear Reviewer fSiU:
> > > > >
> > > > > We thank you for your further comments and would like to address your remaining concerns.
> > > > >
> > > > > To clarify, **in this paper our main contribution is to propose an effective and efficient *pipeline* to improve the stability of hand-object pose estimation via physics simulation, and is not about a specific design of the network or use of the simulator**. The chosen network and simulator are justified in the ablation study to be sufficiently accurate *for the purpose of the aiming task*, and can successfully facilitate the base network to generate more stable and realistic hand-object poses. Meanwhile, our method is not restricted to the current use of the network architecture or physics simulator, and we do agree that that improved designs in future works can better enhance the overall performance.
> > > > >
> > > > > Furthermore, the idea of regressing the stability score can be extended to many practical applications, including the human pose and motion estimation when modeling human-ground interaction. However, we agree that when extending our method to other use cases, the physics metrics, network architecture and simulator may need to be correspondingly modified in order to meet *task-specific requirements*. Nevertheless, for these future works **our main contribution, *i.e.* the overall pipeline and the training strategy can still be beneficial, which reveals the future value of our paper.**
> > > > >
> > > > > We also thank the reviewer for the comment about Eq.(2) and agree that a more precise term should be *semi-implicit* instead.

---

### Official Review · Reviewer_bTxd · 2023-07-04

**Soundness:** 3 good
**Presentation:** 4 excellent
**Contribution:** 3 good
**Rating:** 6
**Confidence:** 4

**Summary:**

This work proposes using an external physics simulation to aid in monocular joint 3D hand and object pose estimation. By analyzing the stability of a perceived grasp inside the simulation, the proposed models learn to factor in grasp dynamics when estimating hand and object pose, producing stable and physically plausible grasps. To circumvent the problem of non-differentiable simulation, a DeepSim model is used as a proxy for learning the dynamics of the physics simulator and enables gradients to be propagated through. Quantitative and qualitative results show that the proposed method produces state-of-the-art results in pose estimation while improving physical realism.

**Strengths:**

1. Leveraging the laws of physics effectively can benefit vision-based systems by providing physical prior. However, the effective use of physical laws and/or simulation is difficult as it creates extra overhead which may lead to intractable systems. This paper provides an effective hand/object pose estimation method that intelligently leverages a (simplified) world model (DeepSim) to learn physical prior from simulation. I find the formulation intuitive, and since the ending estimation pipeline is no longer reliant on the simulator, the pipeline is efficient. Essentially, the network is trained in a physics-aware fashion through the use of the world model.
    1. I think the simplified world model (DeepSim) formulation is interesting and effective. By directly estimation the stability loss the model is easier to learn and can be directly used to optimize the objective.
2. The motivation of the paper is clear, and the proposed solution and components solve the raised issues. The experiments on the analytical gradient and numerical gradient show the necessity of learning the DeepSim model and provide a clear view of the limitation of the current differentiable physics simulator (namely, unstable gradient when dealing with complex contact geometries).
3. Results show that the learned pose and hand estimator outperform SOTA methods in terms of physical plausibility and are comparable in terms of pose estimation accuracy. Qualitative results and simulation videos also show that the proposed method is effective in estimating physically stable grasps.
4. The proposed stability analysis is general and could be applied to other base pose estimation networks as well as other domains such as stable human pose estimation.

**Weaknesses:**

1. I find the applied adhesion force a little questionable. In the real world, humans grasp objects by applying forces, which is akin to small penetration. Applying an adhesion force is similar to having extra suction cups on the fingertips, which is not realistic. How is the force modeled? Is it a constant or is it a function of the contact forces like static friction?
2. The current formulation essentially biases the model towards firm grasps and static holdings of objects. All of the examples shown in the results are grasps that require almost all fingers. How about when the object is resting on the palm or not requiring the support of the hand? Since no ground or tabletop is modeled, would the model also provide grasping when the object is resting on the table? In that case, the object is supported by other forces and fingers do not need to apply to grasp. Would the model still bias toward a solution where all fingers are closed in on the object?
3. As dexterous manipulation is a study of motion, the current setup can be quite limited in modeling faster motion and movement of the objects. Similar to 2, the method is biased toward static and firm grasp, which is not always true when handling an object. The object's own momentum and movement can have a large effect on its stability, which a single-frame model would not factor in.

**Questions:**

I would like to see some discussion on how he adhesion force is applied and modeled, and how well does the model handle non-grasp poses.

---
After rebuttal, my question about adhesion force and some other details are addressed. I would like to maintain a positive rating of this work.

---

**Limitations:**

See weakness. I think the method is biased toward stable grasp and would not be able to model hand and object motion (as opposed to pose).

---

> ### Author Rebuttal · Authors · 2023-08-07
>
> Dear Reviewer bTxd:
>
> We thank you for providing valuable feedback and acknowledging the strength of our work. We hope the below responses can address your concerns.
>
> ### 1. Modeling Adhesion Force
> In practice, we find applying hand-object penetration to emulate the effects of joint torques can degrade the simulation realism as it often encourages the model biasing towards deeper penetration, also observed by [52]. In this way, stability of contact can be achieved by having deep but balanced penetration over multiple sides of the object, or even penetrating through the object, considering the limitation of the penetration resolving in modern physics simulators. **Such artifacts are undesired in the applications and violate physics realism, therefore we choose not to model the interaction via hand-object penetration**.
>
> For the details of adhesion force modeling, we investigate the training data and **carefully adjust the strength of the adhesion force so that a simple touch can not form a stable grasp**. The overall adhesion force strength depends on the number of hand-object contact, where for each contact point, the adhesion force is applied in the direction of contact normal with a fixed strength. The effect of the adhesion force can be observed in the supplementary video, which demonstrates that it reasonably simulates with real world physics **without the need of undesired hand-object penetration**. **Overall, we find it as a better model to emulate the effects of joint torques when the hand is required to remain static during simulation**.
>
> ### 2. Bias Towards Grasping
> Our method does not bias towards grasping or specific forms of grasping as it is additionally conditioned on the input image and supervised with accuracy losses. We mentioned in L219 of the main paper that we include training samples whose ground truth poses are both stable and unstable, *e.g.* with no hand-object interaction, and only impose the stability loss on samples whose ground truth poses are stable. For samples whose ground truth poses are unstable, we train on them using only the accuracy losses. Therefore, our method avoids blindingly producing firm grasping when no hand-object interaction is indicated from the input image. **Please also refer to the Fig.1 of the PDF in the general responses for qualitative results on such cases**.
>
> In addition, we make no assumptions on the contacting fingers and forms of grasping, but rely on the simulator to evaluate the effect of contact on *all detected contacting vertices*. Specifically, unlike [52] that only studies contact on predefined anchors, we consider contact forces on the entire hand mesh for all contact points detected by the simulator in collision detection. This allows us to explore various forms of contact other than grasping with all fingers. **We include more qualitative results in Fig.1 in the PDF of general responses to show that our method can generalize to various forms of grasping, *e.g.* resting on the palm or interacting with a few fingers**.
>
> ### 3. Generalize to Fast Object Motion
> In this work, we address the task of estimating hand and object poses from only a *single image input*, **which is already a challenging task**. Since we do not have information about the object's previous states, from the training data we make a reasonable assumption that the movement is slow and the internal object acceleration is negligible compared to gravity. However, **our method can be easily extended to estimating the hand-object motion from a sequence of input frames with various object initial states**. First, the object velocity $\dot{\mathbf{q}}_0$ can be adjusted to appropriate values if the object does not start to be static. Besides, if the object has its own internal driven force, *e.g.* equipped with a motor, the Eq.(2) can be correspondingly modified to take into the account of additional sources of forces. We believe these extensions are beyond the scope of our work and are suitable for future research.

---

> > ### Comment · Reviewer_bTxd · 2023-08-16
> > **Reviewer Response**
> >
> > I thank the authors for the detailed response.
> >
> > My concerns about "Bias Towards Grasping" and "Generalize to Fast Object Motion" has been addressed. The only remaining question is still centered on the adhesion force. What is the "gain" (L259) in the context of adhesion force? Is the adhesion force also applied to other SOTA methods in the supplementary video?

---

> > > ### Author Response · Authors · 2023-08-17
> > >
> > > Dear Reviewer bTxd:
> > >
> > > We thank you for your comments on our rebuttal. For your remaining questions, the *gain* is a simulator-specific parameter used in the MuJoCo solver to scale the effect of the adhesion force. We describe it for completeness so that our simulation process can be reproduced with the same setting. In addition, we apply the same physics models and simulator, *i.e.* including the application of the adhesion force, for all compared methods in both quantitative and qualitative (including the supplementary video) evaluation for a fair comparison.

---

### Official Review · Reviewer_gag8 · 2023-07-05

**Soundness:** 3 good
**Presentation:** 3 good
**Contribution:** 2 fair
**Rating:** 5
**Confidence:** 3

**Summary:**

This manuscript presents a new approach for generating physically realistic estimates of hand and object pose during hand-object interaction. They argue that a weakness of existing approaches is  that while they utilize approaches to avoid hand-object penetration or enforce contact, they do not enforce physically realistic contact that would obey e.g. gravitational forces, which they term dynamical constraints. Their approach is based on the assumption that the hand-object should be stable, that is, if physical forces were applied to an observed frame for a period of a few hundred milliseconds, the hand and object should show minimal displacement. First, following estimation of hand pose and mesh parameters and object rotation using existing approaches, they use a physics simulator to forecast displacements of the hand-object configuration forward 200 ms, and estimate a stability loss equal to the L2 norm of the displacement. They argue that the gradient of this loss relative to state in the physics simulator is unstable, and therefore they use a separate MLP they call DeepSim to approximate the stability loss, taking as input a concatenation of the hand and object signed distance field and hand-object configuration. They alternately train DeepSim and the base network.

They compare their DeepSim approach on two benchmark datasets with state of the art techniques and achieve competitive, albeit weaker performance on hand and object estimation metrics, but improved performance on a set of more physical criteria – the penetration depth, distance of object displacement and fraction of frames with contact. They motivate the chosen DeepSim architecture through ablations, and show decreases in the magnitude and variance of the gradient using the appproximation approach.

Overall I found the manuscript fairly complete, although of a fairly narrow scope that may mean it is more suited to a specialized computer vision conference. I haven concerns about some of the assumptions made about the physical environment and number of evaluations performed. If some of these questions are addressed I would consider moving up the score.


**Strengths:**

•	The manuscript is well motivated, clearly explained, and well contextualized within the field. There are evaluations across multiple datasets, set of ablations. The conclusions for the most part match the results. This should be published somewhere.
•	The architecture and approach are novel to me, and the approach of having a network learn to smooth the state gradient of a physics simulator could be more broadly useful.
•	The qualitative examples are quite compelling and there are seemingly significant increases in the physics scores for the hand-object interactions.


**Weaknesses:**

•	The method does not achieve state of the art in hand and object pose performance on the chosen datasets.
•	While the approach of smoothing the state gradient could be quite general, the application is narrow in scope. Not only is it limited to hand-object interactions, but the subset of interactions where the grasp is stable. It would be a bit awkward to use this method in practice since it is restricted to the subset of stably grasped frames and those would have to be pre-detected.
•	There are assumptions about contact force strength and the orientation of the gravity axis that are made and it is not clear to what degree they impact the results.


**Questions:**

•	How were hyperparameters for the losses chosen?
•	Can you provide accuracy comparison values (eg MJE) for the chosen examples? It would be nice to see if they are representative. Alternatively, it would help to see examples of what the failure mode of the techniques were.
•	Why are results from only a single random seed presented?
•	L258 “We set the gravity acceleration as 9.8 m/s2  in the y direction of the camera frame. For the adhesion force, we empirically set the gain as 100 and the force as 10N, so that a simple touch is not sufficient to stably grasp the object. “ To what degree is it accurate to assume the gravity axis in these datasets is aligned with the images? To what degree do slight variations in the gravity axis produce large changes in the object pose? Does this affect the comparison between techniques in Tables 1 and 2.
•	It is disappointing that the hand ground truth is not available in Table 2 and it makes it difficult to evaluate the technique as there is really only a single dataset with joint errors. Are there other splits or datasets that could be used?
•	Can you report the performance of networks trained with NimblePhysics and FiniteDifference gradients? It is unclear, especially with the Finite Difference network, if the networks simply don’t converge well or if there is a more catastrophic reason why they cannot be used for training. This was a key motivation for the DeepSim approach and it would be nice to verify that they are problematic. It would be interesting if they produced e.g. more physically plausible results.


**Limitations:**

Yes.

---

> ### Author Rebuttal · Authors · 2023-08-07
>
> Dear Reviewer gag8:
>
> We thank you for providing valuable feedback and acknowledging the strength of our work. We hope the below response can address your concerns.
>
> ### 1. Concerns about Performance in Accuracy
> Please refer to the general responses #1. for the discussion about the accuracy performance.
>
> ### 2. Scope of Application
> **Our proposed method is a general framework and works with any task-specific base networks and physics models, therefore is not restricted in hand-object interaction only**. For instance, other tasks including human pose [B3] and motion estimation [B2] also require the results to be physically stable under the interaction to the ground and scenes. Our methods can be easily extended to these tasks with modified base networks and physics simulators, using the DeepSim network to smoothly connect the two parts. This is also acknowledged by the reviewer `bTxd` in strength 4.
>
> **Besides, our method is also not restricted to stable grasping frames only since we train the model jointly using accuracy and stability losses**. We mentioned in L219 of the main paper that during training, we include all samples whose ground truth are both stable and unstable, *e.g.* having no hand-object contact. To avoid biasing towards grasping, we mask out the stability loss for samples whose ground truth is not stable, and train on these samples using only accuracy losses. Therefore, our method can generalize to both types of test samples. We include more results on non-grasping samples in the Fig.1 of the PDF in the general responses for illustration. **In summary, since our model is conditioned on the input image, it will not blindly predict all samples as having a grasping hand if the image does not indicate so**.
>
> ### 3. Impact of Design Choices
> We make assumptions about the contact force strength and the direction of gravity based on the observation of the training data. We empirically find these design choices reasonably align with real world physics. Please refer to the supplementary video for qualitative evaluation.
>
> Since in practice, it is difficult to perfectly simulate real world physics, we take special care in training to avoid the model being misled by imperfect simulation. Specifically, we examine all training data and *do not* impose stability loss on samples whose ground truth pose are *unstable under our assumption*. For firm grasping cases and non-contacting cases, they remain stable/unstable regardless of the gravity direction. For other samples, only those that align with our assumption will be included for training with stability loss. **Consequently, the impact is restricted mainly to the reduction of training data**.
>
> In testing, we also ensure that the selected samples follow our assumption (see the GT in Table 1 of the main paper) or manually verified by the previous work [52]. Hence the physics metrics shown in Table 1 & 2 are meaningful and comparable.
>
> ### 4. Hyper-parameters in Losses
> We follow the same training pipeline as [32] to set the same weights for accuracy losses. For the stability loss, we set the weight as $\lambda_s = 0.1$ so that all losses are roughly in the same scale.
>
> ### 5. Accuracy Comparison for Selected Examples
> Please refer to Table 4 of the PDF in general responses for the accuracy of selected examples in the main paper.
>
> ### 6. Analysis of Failure Cases
> Please refer to Fig. 2 of the PDF in general responses for the analysis of the failure case.
>
> ### 7. Random Seed
> We show it in Table 3 of the PDF in general responses for the results of multiple runs on the two datasets. The result shows a low variance in our method and indicates a stable performance of our model.
>
> ### 8. More Quantitative Evaluation
> We further follow [52] to compare on the HO3Dv1 split for more quantitative evaluation. We adopt the same setting as [52] for a fair comparison. Please refer to Table 1 of the PDF in general responses for the result. Note that both [19, 52] are state-of-the-art methods that explicitly adopt physics priors when modeling hand-object interaction. Compared to them, our method achieves consistently better performance in both accuracy and stability.
>
> ### 9. Results for NimblePhysics and FiniteDifference
> As shown in the Fig. 4(a) of the main paper, both methods fail to converge as the training losses can not be decreased, **leading to significantly worse performance and out-of-range scores**. For FiniteDifference (green curve), we can clearly observe an increase of the loss as training progresses, indicating a failure of training. This is because it produces incorrectly large gradient due to the sudden velocity change during penetration resolving. For NimblePhysics, the noisy gradient, as demonstrated in Fig.4 (b), also prevents the loss from decreasing, which illustrates the necessity of our method in smoothing the gradient. In addition, we mentioned in L37 of the supplementary material that the NimblePhysics takes around 120 hours to train a single epoch, as computing the state gradient for complex contact geometry is computationally expensive. **In consequence, it is also intractable to apply it when training on large-scale datasets**.
>
> ### Bibliography
> [B2] Gärtner E, Andriluka M, Coumans E, et al. Differentiable dynamics for articulated 3d human motion reconstruction[C]//Proceedings of the IEEE/CVF Conference on Computer Vision and Pattern Recognition. 2022: 13190-13200.
>
> [B3] Tripathi S, Müller L, Huang C H P, et al. 3D human pose estimation via intuitive physics[C]//Proceedings of the IEEE/CVF Conference on Computer Vision and Pattern Recognition. 2023: 4713-4725.

---

> > ### Comment · Reviewer_gag8 · 2023-08-13
> > **Thank you, slightly raising the scores**
> >
> > I appreciate the authors' response. I like this paper and while I acknowledge that the use of physics simulators is not entirely novel, I do think there is a lot of value in the current approach. Because of this I am going to slightly bump my score, but I would not strongly advocate for acceptance.
> >
> >  I am somewhat less concerned on a re-read about the method not hitting SOTA for object and pose keypoint detection but I also think the physics metrics could be presented in a realistic use case to be more convincing.
> >
> > I don't feel like two of my points are really addressed. I don't see how this method will apply to cases where the stability levels are unclear , eg during walking phases where your heel is off the ground higher stability can be achieved by increasing surface area. This is also noted by bTxd. I also don't see sufficient consideration of just guessing the gravity axis.

---

> > > ### Author Response · Authors · 2023-08-16
> > >
> > >
> > > Dear Reviewer gag8:
> > >
> > > We thank you for your kind support of raising the scores and further comments about this paper. We hope the below responses can address your remaining concerns.
> > >
> > > We acknowledge that the simulation process may not perfectly align with real world physics as we infer from only single image input. This can inspire our future research direction to exploit additional knowledge, such as temporal information from video inputs or statistical analysis of human physics, to better conform to reality. Nevertheless, we would like to highlight that our method can generalize to other use cases with modified physics models and simulators, **without relying on oversimplified rule-based heuristics or affecting the design of the DeepSim network**. For instance, we can follow the design of physics models and the simulator in other works like [B2] when modeling stability in human-ground interaction.
> > >
> > >
> > > In terms of the gravity axis, we acknowledge that we assume the gravity direction is known. In future works, we can calibrate a more precise gravity direction from the image observation by further exploiting semantics and normal map information.
> > >
> > >
> > > [B2] Gärtner E, Andriluka M, Coumans E, et al. Differentiable dynamics for articulated 3d human motion reconstruction[C]//Proceedings of the IEEE/CVF Conference on Computer Vision and Pattern Recognition. 2022: 13190-13200.

---

### Official Review · Reviewer_oW7V · 2023-07-06

**Soundness:** 3 good
**Presentation:** 4 excellent
**Contribution:** 3 good
**Rating:** 5
**Confidence:** 3

**Summary:**

This work presents a novel pipeline for 3D hand-object pose estimation, focusing on improving arbitrary base hand/object pose estimators with applying physical simulation and its induced physical loss (fitted as a neural network). The performance is tested on DexYCB and HO3D data sets and the method is compared against many SotA baselines. Results show improved performance mostly on the physics metrics.

**Strengths:**

- The proposed idea and method are reasonable, interesting, and valid.
- The performance on the introduced physics metrics gets improved over baseline methods.
- The paper writing is good and the paper is easy to read and follow.
- Qualitatively, the proposed method does generate more physically realistic hand/object interaction poses.

**Weaknesses:**

- My major concern is that the performance regarding hand/object pose estimations is worse than the baselines [32, 52] in Table 1 & 2. Since the proposed method is particularly emphasized on optimizing physical realism, I think it's under the expectation that the method performs better in terms of the proposed physical metrics. But the whole point of doing this should be to improve the hand/object pose estimation results, which is not achieved as shown in the tables.
- I'm confused by Fig. 1. The gt object pose looks quite different from the input image. Is this the dataset annotation issue or any visualization issue? Why is the gt object pose annotated so off? It's hard to say it's the problem of the baseline [32] if the gt is so off.
- Why the related work [47, 24] are not compared in Table 1 / 2? I think it's important to compare against them as they also considered physical realism for the same task and the authors have discussed them in related work that they have disadvantages. It's better to show them in numbers.
- In additional, I feel the idea of using physical simulation as loss to supervise interaction-oriented perception tasks is not new, as the many papers cited by the authors in the related work section and more others.

**Questions:**

see weakness

**Limitations:**

no issue found

---

> ### Author Rebuttal · Authors · 2023-08-06
>
> Dear Reviewer oW7V:
>
> We thank you for providing valuable feedback and acknowledging the strength of our work. We hope the below responses can address your concerns.
>
> ### 1. Concerns about Accuracy Performance
> We hypothesize that the comment intends to mean that our method performs worse than the baseline method [32, 48] instead of [32, 52]. In Table 2 of the main paper we show that our method achieves superior performance in both accuracy and stability compared to [52], which also explicitly enforces physics realism in hand-object pose estimation. Our accuracy is comparable to [32, 48] even though they use more augmented data during training, and our method achieves superior results when the baseline methods are trained with the same amount of data. **Please refer to the general responses #1. for detailed discussion related to accuracy performance.**
>
> ### 2. Clarification for Fig.1
> We believe that there are some misinterpretations for Fig.1. In Fig.1, the visualization for the ground truth annotation refers to the figure in the *first row*, second column, instead of the second row, second column, *i.e.* caption at the *bottom* of the figure. The figure in the second row, second column instead visualizes the hand-object pose in a rotated view angle in order to better demonstrate that our method generates stable contact in the occluded area. **In the correct figure for visualizing the ground truth annotation, we project the ground truth hand and object mesh in the image space and prove that they align well with the input image, hence there should be no significant errors in the annotation.**
>
> ### 3. Comparison to [47, 24]
> Both related works [47, 24] do not release codes for evaluation. [47] also does not release the physics models and exact settings used in simulation. We are therefore unable to reproduce their works for evaluation. Besides, both [47, 24] are evaluated on self-collected small-scale datasets, which are also not publicly available. Hence we are unable to evaluate our method on their datasets and compare with their performance. Nevertheless, in quantitative comparison, we emphasize the comparison with [52], which is a recent work that also enforces physical realism in hand-object pose estimation. Table 2 of the main paper shows that our method consistently outperforms [52] in terms of both accuracy and stability.
>
> ### 4. Clarification for Contribution and Novelty
> **Our main contribution is not about being the first to integrate physics simulator or simulator-induced losses in the refinement pipeline, but proposing a more *effective* and *efficient* method that learns from physics simulation for estimating stable hand-object pose with complex contact geometry**. Due to the intrinsic discontinuity in simulation process and resulting noisy state gradient, directly imposing losses from differentiable physics simulators and refining the base network via gradient descent is challenging. To this end, previous works perform brute-force search [47] over limited configuration space, or rely on additional global optimization [B2]. **Such test-time optimization strategy is computationally expensive and therefore has limited application**. Alternatively, other works [39, 11] propose to adopt a deep reinforcement learning framework to work with non-differentiable simulators. However, **these methods do not generalize to unseen data and are also difficult to converge in training**. In contrast, we propose to adopt a neural network  DeepSim that can smoothly approximate state gradient from the simulator and effectively refine the base network via back-propagation, **which is not addressed by related works**. Besides, since our method does not require test-time optimization, more stable results can be produced via a simple forward call of the refined base network, which is more efficient and practically applicable compared to previous works. Ablation study in **Section 4.5 of the main paper also demonstrates the effectiveness and superiority of our method compared to directly using the physics simulator as supervision**.
>
> Furthermore, We refer the reviewer to the general responses #2. for the novelty of our method compared to other works that adopt neural networks for simulation approximation.
>
>
> ### Bibliography
> [B2] Gärtner E, Andriluka M, Coumans E, et al. Differentiable dynamics for articulated 3d human motion reconstruction[C]//Proceedings of the IEEE/CVF Conference on Computer Vision and Pattern Recognition. 2022: 13190-13200.

---

> > ### Comment · Reviewer_oW7V · 2023-08-22
> >
> > Thank you for the rebuttal. I've raised up my score to Borderline Accept.

---

### Official Review · Reviewer_HRq6 · 2023-07-10

**Soundness:** 3 good
**Presentation:** 3 good
**Contribution:** 3 good
**Rating:** 5
**Confidence:** 4

**Summary:**

Authors propose an approach for 3D pose estimation for hand-object interaction from a single image. Unlike prior work which is purely data driven (except for some works in robotics literature) and focus on visual quality, this work aims to also grasp stability. To this end, authors use physics based simulation to model grasp stability. Since PBS is typically non-differentiable, authors propose to train a neural network (with full supervision from the simulation) to emulate the simulation. This allows them to differentiably approximate the simulation.
Authors outperform baselines on DexYCB and HO3D datasets.

**Strengths:**

+ Most existing works focus on visual plausibility. It is interesting to see this work also reason about physical stability.
+ The idea of approximating non-differentiable simulation has previously been used () but it is still under explored. It would be better if authors provide more context about prior work in this space.
+ Authors outperform the baselines.

**Weaknesses:**

[Technical]
1. Eq. 2: Can the authors provide a bit more motivation/intuition around the equation. What does it do? Why is it correct? Is it a general formulation or does it have specific applications. Current reference ([14]) is a 650+ page book on Classical Mechanics, which does not help the reader much. Maybe point to specific portion of the text that elaborates on the equation.
It is not clear how did the authors arrive at this equation from Euler Lagrange equation.
This derivations should be included in supp. mat. at least.

2. Sec 3.3: Approximating non-differentiable functions with a neural network is a well known problem. The most straightforward solution is to train a MLP with supervised data. Isn’t this exactly what DeepSim is doing?
This can be traced back to “Approximation of functions and their derivatives: A neural network implementation with applications”, Nguyen-Thien et. al. Applied Mathematical Modelling, 1999
Other domains such as cloth simulation have also used similar techniques to train a neural network to predict the outcome of a physical simulation. (Holden et al. Eurographics’19)
Can the authors elaborate how is their proposed DeepSim different? This is an under explored area so it is okay to have some similarities with prior work but authors should provide more context and flesh out what are the new insights here.

3. L142: How are the physical properties of the hand and object eg. mass, coefficient of friction etc. obtained from the input image for simulating forces? Does this generalise?

4. Eq. 7: Do we need to mark 8 corners on all template meshes? Isn't this restrictive? Why not put the loss on object mesh vertices directly?

5. Eq. 3: What about rotation? A grasp is still not stable if the object rotates due to a static grasp? Why consider only translation?

6. L140: If M is a matrix, what does M(\cdot) mean?

[Minor]
- Missing related work:
TOCH. Zhou et al. ECCV’22. They learn to predict stable grasps from data.
Jiang et al. ICCV’21: They learn to predict stable contacts on object and synthesise hand to match the contacts.

**Questions:**

Some key formulations are unclear to me (see pt.1) this is important to clarify as it is one of the main contributions of the work.
More clarity around design choices would also help the manuscript (see pts. 2-6).

**Limitations:**

Authors discuss potential limitations and broader impact in the paper.

---

> ### Author Rebuttal · Authors · 2023-08-06
>
> Dear Reviewer HRq6:
>
> We thank you for providing valuable feedback and acknowledging the strength of our work. We hope the below responses can address your concerns.
>
> ### 1. & 6. Clarification of Eq.(2)
> For the notation, $\mathbf{M}(\mathbf{q}_t)$ denotes that the object inertia matrix $\mathbf{M}$ is determined by the object configuration $\mathbf{q}_t$, where the parenthesis $(\cdot)$ indicates the dependency. Other quantities in Eq.(2) follow the same convention. The notation is commonly used in related literature [44, 50]. In addition, we wish to correct a typo that the left-hand side of the first equation in Eq.(2) should be $\mathbf{M}(\mathbf{q}_t)$ instead of $\mathbf{M}(\mathbf{q}\_{t+1})$. We apologize for the confusion caused in understanding the equation and will revise it in the final version.
>
> To elaborate further, Eq.(2) defines how the object state, including the configuration $\mathbf{q}_t$ and velocity $\dot{\mathbf{q}}_t$, is calculated and updated at each simulation time $t$. Specifically, we formulate the first equation of Eq.(2) to state that the system momentum changes due to the corresponding impulses, *i.e.* $\mathbf{M}\dot{\mathbf{q}}\_{t+1} = \mathbf{M}\dot{\mathbf{q}}_t + \mathbf{f}\Delta t$, where $\mathbf{f}$ consists of the gravitational and Coriolis force $\mathbf{c}$ as well as contact-induced forces $\mathbf{f}_C, \mathbf{f}_A$. We use the above Lagrangian dynamics equation in order to work in generalized coordinates. Once we have solved the updated velocity $\dot{\mathbf{q}}\_{t+1}$, we can then compute the updated configuration $\mathbf{q}\_{t+1}$ using the discrete time Euler integration scheme, as defined in the second equation. The current citation [14], also co-cited in [B1], refers to the definition of the general Lagrangian dynamics equation for rigid bodies (Chapter 5), while the first equation of Eq.(2) is a specific instantiation of it in order to introduce relevant forces in $\mathbf{f}$ for our task. This equation is implemented by the MuJoCo simulator [44] used in our work. We will include more references [B1,44] to provide a better clarification on this equation.
>
> ### 2. Difference to Network-based Simulation Approximation
> Please refer to the general responses #2. for discussion of our contribution and the difference to other works that use neural networks to approximate physics simulation.
>
> ### 3. Obtaining Physics Properties
> We set the object physics properties based on the previous annotations (See footnote in page 2, supplementary materials). Since the hand remains static during simulation, the hand mass and inertia is irrelevant and we set them as constants in a similar scale to the object. For all physics coefficients, *e.g.* coefficient of friction, we use default values in MuJoCo, which are optimized to be particularly suitable and generalizable for simulating common objects. We manually verified that all simulation parameters reasonably align with real world physics, where the effects are demonstrated in the supplementary video. However, we mentioned in our limitation that the simulation settings may not be perfect for objects with complex and rare physics properties, which is best to be modified for task-specific requirements. Note that we do not estimate physics properties from input images.
>
> ### 4. Using Corner Losses
> We follow the same training pipeline as the baseline method [32] to use object corner losses for a fair comparison. Specifically, we compute the 8 object corners from the ground truth mesh by taking the max and min values along each of the 3 axes, resulting in 8 vertices of the tightest object bounding box. No manual annotation of these corners is required. Since we are refining the 6 DoF object pose as the base network output instead of individual vertex positions, training with corner losses should be equally effective in principle and computationally more efficient than vertex loss. It also better supports batched training when objects have different numbers of vertices.
>
> ### 5. Using Rotation in Stability Loss
> We design the stability loss using only object center displacement after simulation as it is empirically sufficient to determine the stability of the estimated pose. In the implementation, since we simulate for a relatively long time (T = 200ms) and set a small threshold (displacement less than 1cm) to classify as being stable, we empirically observe that samples with a large object rotation change after simulation also tend to exceeds the displacement threshold, which can be correctly classified as unstable.
>
> In addition, We show in Table 3 of the main paper that the model MLP + RT (using both rotation and translation), performs worse than MLP + T (translation only) due to the increasing difficulty of regression with the DeepSim. We therefore choose to consider object translation only.
>
> ### [Minor] Other Related Works
> We thank the reviewer for providing other related works. However, both suggested papers focus on generating or refining contact given *3D* inputs, which is not directly comparable to our methods as we estimate poses from *image* inputs. For completeness, we will include the discussion of these papers in Section 2 in the final version.
>
> ### Bibliography
> [B1] Andrews S, Erleben K, Ferguson Z. Contact and friction simulation for computer graphics[M]//ACM SIGGRAPH 2022 Courses. 2022: 1-172.

---

> > ### Comment · Reviewer_HRq6 · 2023-08-16
> > **Post rebuttal update**
> >
> > Thanks authors for the rebuttal. It clarified my doubts. After reading other reviews and rebuttals, I maintain my positive overview of the work. The the discussion around integrating rotation in the stability loss is interesting and can be briefly incorporated in the "limitations/ future works" section.

---

> > > ### Author Response · Authors · 2023-08-16
> > >
> > > Dear Reviewer HRq6:
> > >
> > > We thank you for maintaining positive overview of the work. We will further discuss the design of the stability loss in the final version.

---

### Author Rebuttal · Authors · 2023-08-05

We thank all the reviewers for their time and valuable comments. Below, we first clarify common concerns raised by multiple reviewers.

### 1. Concerns about Accuracy Performance (Reviewer oW7V, gag8)
In this paper, we aim to address the task of estimating stable hand and object poses from single-image inputs, which is **important for applications that demand *robust* hand-object interaction**. For instance, in dexterous manipulation, ensuring successful grasp and manipulation of the target object often takes precedence over precisely replicating the exact contact points. However, previous learning-based methods such as [32, 48] often produce suboptimal results where although the hand fingers are close to the ground truth contact positions, a stable grasp is not actually formed. In consequence, while they exhibit higher estimation accuracy, i.e. with reduced hand and object pose errors, they are unsuitable for these applications. To this end, we place a stronger emphasis on physics metrics to better cater to the requirements of such applications.

**Compared to previous methods [52, 19] that also explicitly optimize for physics realism, our method achieves state-of-the-art performance in both accuracy and stability**. In particular, [52, 19] impose over-simplified assumptions on contact dynamics, in contrast, we learn complete dynamics priors from the physics simulator and thus avoid biasing towards a restricted set of stable poses. We also highlight that our methods are more efficient compared to [52, 19] since no test-time optimization is needed.

We note that baseline methods [32, 48] have a higher accuracy performance, this is because we evaluate them using their officially released model weights, which are **trained with significantly more augmented data**. In particular, they synthesize additional hand and object images and corresponding annotations from various rotated views to mitigate occlusion ambiguity. However, since rotating the hand and object poses may alter the stability status of the original configuration, we did not apply the same augmentation strategy and only used a reduced amount of data for the training with the stability loss. Nevertheless, our method still achieves comparable accuracy thanks to the dynamics priors learned from the physics simulation. Qualitative results also demonstrate that estimated hand and object poses visually align with input images. **In Table 4 of the attached PDF, we show that our method achieves better accuracy than [32] (which uses the same base network with us) when training with the same amount of data, proving that the accuracy of our method is comparable**. Furthermore, the evaluation also proves that having a higher accuracy performance does not necessarily result in better stability, which jusitfies the motivation of our work.

### 2. Difference to Methods using Neural Networks to Approximate Simulator (Reviewer HRq6, oW7V, fSiU)
Unlike previous methods [42, 37] that attempt to directly approximate the entire physics simulator by regressing *complete simulated states*, **our key insight in designing the DeepSim is to regress the *scalar* stability loss supervised by the simulator, which is a simplified task and practically more feasible to achieve**. This improved design allows the DeepSim network to accurately infer the pose stability and better generalize to unseen test data, as proved in the ablation study, *i.e.* Table 3 of the main paper.

To elaborate the motivation, our goal during physics refinement is to quantitatively evaluate the stability of the hand and object pose estimated from a base network and refine the estimation stability. While modern physics simulators can serve as such an evaluator, due to the *complex contact geometry* and resulting noisy state gradient, the stability loss evaluated from the simulator can not be directly back-propagated to refine the base network. We therefore propose the DeepSim network to learn from the simulator, *i.e.* asked to replicate the same evaluated stability supervised by the simulator, *instead of being itself as a simulator*, while preserving smooth gradient that are suitable for back-propagation. We believe the design of the DeepSim shares more similarity to the *discriminator* in the GAN [15], however, it has no adversarial relationship to the base network (akin to the generator) and thus is much easier to train.

---

### Decision · Program_Chairs · 2023-09-21

**Decision:**

Accept (poster)

**Comment:**

The paper presents a physics-aware solution for 3D hand-object pose estimation and originally obtains borderline to positive scores. The rebuttal has successfully addressed some of the reviewers' concerns. All reviewers gave positive scores in the end. The ACs concurred with the reviewers' ratings and chose to accept the paper. The authors are strongly encouraged to fulfill their promise of making necessary content edits when preparing the final version.